# PassNet: Learning pass probability surfaces from single-location labels. An architecture for visually-interpretable soccer analytics

## Abstract

We propose a fully convolutional network architecture that is able to estimate a full surface of pass probabilities from single-location labels derived from high frequency spatio-temporal data of professional soccer matches. The network is able to perform remarkably well from low-level inputs by learning a feature hierarchy that produces predictions at different sampling levels that are merged together to preserve both coarse and fine detail. Our approach presents an extreme case of weakly supervised learning where there is just a single pixel correspondence between ground-truth outcomes and the predicted probability map. By providing not just an accurate evaluation of observed events but also a visual interpretation of the results of other potential actions, our approach opens the door for spatio-temporal decision-making analysis, an as-yet little-explored area in sports. Our proposed deep learning architecture can be easily adapted to solve many other related problems in sports analytics; we demonstrate this by extending the network to learn to estimate pass-selection likelihood.

## 1 Introduction

Sports analytics is a fast-growing research field with a strong focus on data-driven performance analysis from professional athletes and teams. Soccer, as well as many other team-sports, has recently benefited from the availability of high-frequency tracking data of both player and ball locations, facilitating the development of fine-grained spatio-temporal performance metrics (Rein & Memmert, 2016; Stein et al., 2017). One of the main goals of performance analysis is to answer specific questions from soccer coaches, but in order to do so we require models to be robust enough to capture the nuances of a complex sport, and be highly interpretable so findings can be communicated effectively. In other words, we need models to be both accurate and also translatable to soccer coaches and game analysts in visual terms.

The vast majority of research in soccer analytics addresses questions related to either the most commonly observed events during a match, such as goals, shots and passes or the effects of players' movements and match dynamics (Gudmundsson & Horton, 2017). Most modeling approaches share one or more common issues, such as: heavy use of handcrafted features, no visual interpretability, and coarse representations that ignore meaningful spatial relationships. While many of these methods are able to provide valid insights into specific problems, in terms of model design we still lack a comprehensive approach that can learn from lower-level input, exploit spatial relationships on any location (beyond location of the origin and destination of events), and provide not only accurate evaluations of observed events at their given location but also an estimation of unobserved events at any other location on the field.

In this work we present a deep neural network architecture designed to estimate the probability of passing events in soccer from low-level tracking-data input and weak labeling of each event's success. The network is able to estimate the probability of pass success at any location on the field while providing a visual representation of these probabilities. Beyond pass probabilities, such an architecture can be adapted to many other problems such as pass selection, pass risk and reward

evaluation, expected possession value estimation, and player movement evaluation, among many others.

The presented architecture was inspired by recently developed fully convolutional networks that have been proven to be successful for image segmentation (Long et al., 2015; Yu & Koltun, 2015; Pathak et al., 2015; Jing & Tian, 2019). Our problem is similar in many ways to that of image segmentation, where the objective is to provide a pixel-level classification of objects in the image. Image segmentation is usually addressed as a weakly supervised learning problem where only single labels of the objects in the image are provided, instead of pixel-wise labeling. In the case of pass probability, as well as in most of the other mentioned applications in soccer, only a single label is available for each pass to indicate whether it was successful or not, as well as the location of the origin and destination of passes. However, unlike image segmentation where an object label is typically associated with multiple pixels in the image, here the label only corresponds to a single location in the original input. This set up presents an extreme case of weakly supervised learning. We use a *deep jet*-inspired nonlinear feature hierarchy (Long et al., 2015) to combine coarse and fine layers, through convolutional fusion layers. One of the main contributions of this paper is the computation of the network loss by single-pixel masking, according to the destination location associated to each pass event. We show that from backpropagation of the loss between a single-location prediction of the full output matrix and the known outcome of the event we can learn complex spatial features and successfully estimate the pass probability map for any location on the field.

We compare our model with a baseline estimation of pass probability, as well as with linear and non-linear models built on top of handcrafted features, and achieve considerably better results for single pass probability estimation, while also properly estimating the complete probability map. In the following sections we describe related work, provide a detailed explanation of the architecture and design considerations of the model, and present experimental results on a broad set of $208, 489$ passing events from professional soccer matches.

## 2 RELATED WORK

From an applied standpoint our work is related to several other approaches aimed at estimating pass probabilities and other performance metrics derived from spatio-temporal data in soccer. Regarding the technical approach we leverage recent findings on weakly supervised learning problems and the application of fully convolutional neural networks for image segmentation.

**Soccer analytics**    Pass probability estimation has been approached in several ways. A physics-based model of the time it takes each player to reach and control the ball has been used to derive pass probabilities on top of tracking data (Spearman et al., 2017). Other approaches include the use of dominant regions, influenced by motion information, to determine which player is most likely to control the ball after a pass (Gudmundsson & Horton, 2017), or the building of linear models based on a carefully selected set of handcrafted features (Power et al., 2017). The related problem of pass selection has been approached by the application of convolutional neural networks that predict the likelihood of passing to a specific player on the attacking team(Hubáček et al., 2018). The estimation of pass value has been approached either by the experted-guided development of algorithmic rules (Cakmak et al., 2018; Link et al., 2016), the application of standard machine learning algorithms on a set of handcrafted features (Gyarmati & Stanojevic, 2016; Power et al., 2017), or problem-specific deep learning models with dense layers and single output prediction (Wagenaar et al., 2017; Fernández et al., 2019). While some of the related work has estimated probability surfaces by inference on a set of discrete pass destination location (Spearman et al., 2017; Fernández et al., 2019), none has yet approached learning of probability surfaces directly.

**Fully convolutional networks and weakly supervised learning**    Fully convolutional networks have been extensively applied to semantic image segmentation, specifically for the pixel-labeling problem to successfully detect wide pixel areas associated with object in images. The approach most related to our work builds a hierarchy of features at different sampling levels that are merged together to provide segmentation regions that preserve both fine and coarse details (Long et al., 2015). From a

learning perspective image segmentation has been approached either as fully supervised (Long et al., 2015; Noh et al., 2015), weakly supervised (Pathak et al., 2015; Pinheiro & Collobert, 2015; Zhang et al., 2013; Durand et al., 2017), or semi-supervised learning problem (Papandreou et al., 2015; Bai et al., 2017; Souly et al., 2017). Commonly available labels are associated with many other pixels in the original image. However, in our case, labels are only associated with a single location in the desired probability map, making our learning problem a rare and extreme case of weakly supervised learning.

## 3 A DEEP MODEL FOR INTERPRETABLE ANALYSIS IN SOCCER

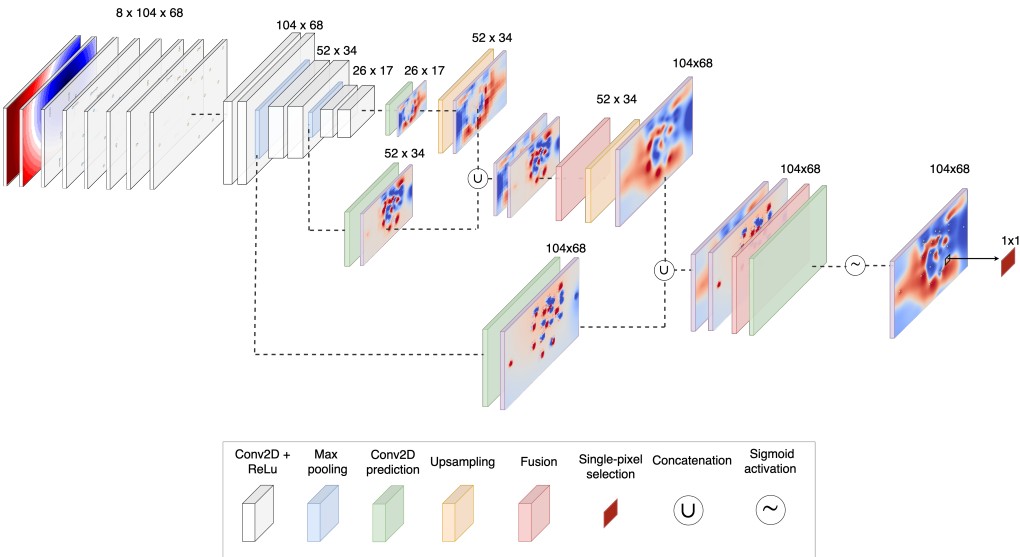

Figure 1: PassNet architecture for a coarse soccer field representation of $104 \times 68$ and 8 input channels.

We build our architecture on top of tracking data extracted from videos of professional soccer matches, which consist of the 2D-location of players and the ball at 10 frames per second, along with manually tagged passes. At every frame where a pass has been labeled we take a snapshot of the tracking data and create a representation of a game situation consisting of a $l \times h \times c$ matrix, where $c$ channels of low-level information are mapped to a $l \times h$ coarse spatial representation of the locations on the field. We seek an architecture that is able to learn both finer features at locations close to a possible passing destination and features considering information on a greater spatial scale. For passes, local features might be associated with the likelihood of nearby team-mates and opponents reaching the destination location and information about local spatial pressure. On the other hand, greater scale features might consider player's density and interceptability of the ball in its path from the location of origin. Finally, we seek to estimate this passing probability to any other location on the $l \times h$ spatial extent of the field.

This representation of the situation is processed by the deep neural network architecture presented in Figure 1. The network creates a feature hierarchy by learning convolutions at $1x$, $1/2x$ and $1/4x$ scales, while preserving the receptive field of the filters. Predictions are produced at each of these scales, and then upsampled nonlinearly and merged together through fusion layers. A sigmoid activation layer is applied to the latest prediction to produce pass probability estimations at every location, preserving the original input scale. During training, a single-location prediction, associated with the destination of a sample pass is selected to compute the log-loss that is backpropagated to adjust the network weights.

### 3.1 Reasoning behind the choice of layers

The network incorporates different types of layers: max pooling, linear, rectilinear linear unit (ReLu) and sigmoid activation layers, and 2D-convolutional filters (conv2d) for feature extraction, prediction, upsampling and fusion. In this section we present a detailed explanation of the reasoning behind the choice of layers and the design of the architecture in general.

**Convolutions for feature extraction**    At each of the $1x$, $1/2x$ and $1/4x$ scales two layers of conv2d filters with a $5 \times 5$ receptive field and stride of 1 are applied, each one followed by a ReLu activation function layer, in order to extract spatial features at every scale. In order to keep the same dimensions after the convolutions we apply a symmetric padding to the input matrix of the convolutional layer. We chose symmetric-padding to avoid border image artifacts that can hinder the predicting ability and visual representation of the model (Wu et al., 2019).

**Fully convolutional network**    There are several conceptual and practical reasons for considering convolutional neural networks (convnets) for this problem. Convolutional filters are specifically designed to consider the relationships between nearby pixels so learned features are spatially aware. Convnets have proven to be successful in data sources with an Euclidean structure, such as images and videos, so a 2D-mapping of soccer field location-based information can be expected to be an ideal data structure for learning meaningful features. Also, these features are expected to be non-trivial and complex. Convnets have been proven to learn what are sometimes more powerful visual features than handcrafted ones, even given large receptive fields and weak label training (Long et al., 2014). Regarding the architecture design, we are interested in learning the full $l \times h$ mapping of passing probabilities covering the whole of a soccer field, for which fully convolutional layers are more appropriate than classical neural networks built for classification, by subtracting dense prediction layers, and using 1x1 convolutions instead.

**Pooling and upsampling**    The network applies downsampling twice through max pooling layers in order to obtain the $1/2x$ and $1/4x$ representations. Since activation field size is kept constant after every downsampling step the network is able to learn filters of a wider spatial extent, leading to the detection of coarse details. We learn non-linear upsampling functions at every upsampling step by first applying a $2x$ nearest neighbor upsampling and then two layers of convolutional filters. The first convolutional layer consists of 32 filters with a $3 \times 3$ activation field and stride 1, followed by a ReLu activation layer. The second layer consists of 1 layer with a $3 \times 3$ activation field and stride 1, followed by a linear activation layer. This upsampling strategy has been shown to provide smoother outputs and to avoid artifacts that can be usually found in the application transposed convolutions (Odena et al., 2016).

**Prediction and fusion layers**    Prediction layers consist of a stack of two convolutional layers, the first with 32 $1 \times 1$ convolutional filters followed by an ReLu activation layer, and the second consisting of a 1 $1 \times 1$ convolutional filter followed by a linear activation layer. Instead of reducing the output to a single prediction value, we keep the spatial dimensions at each step and use $1 \times 1$ convolutions to produce predictions at each location. The stack learns a non-linear prediction on top of the output of convolutional layers. In order to merge the outputs at different scales we concatenate the pair of matrices and pass them through a convolutional layer of 1 $1 \times 1$ filter. In other approaches, where predictions at different scales are merged (Long et al., 2015; Wei et al., 2018), usually matrixes are joined by simple addition. However, this addition would be highly dependent on the distribution of output values at each prediction step, and without proper normalization the differences in numerical ranges might over-emphasize the outputs in one scale over the other. Or fusion layer learns a simple weighting function to add predictions at every location.

### 3.2 Learning from single-location labels

We seek a model that can produce accurate predictions of the pass probability to every location on a $l \times h$ coarsed representation of a soccer field, given a $l \times h \times c$ representation of the game situation at the time a pass is made. In training, we only count with manually labeled location of the pass destination and a binary label of the outcome of the pass.

**Definition 3.1 (PassNet)** *Let $X = \{x | x \in \mathbb{R}^{l \times h \times c}\}$ be the set of possible game state representations at any given time, where $l, h \in \mathbb{N}_1$ are the height and length of a coarse representation of soccer field, and $c \in \mathbb{N}_1$ the number of data channels, a PassNet is a function $f(x; \theta), f : \mathbb{R}^{l \times h \times c} \to \mathbb{R}^{l \times h}_{[0,1]}$ where $f$ produces a pass probability map and $\theta$ are the network parameters.*

**Definition 3.2 (Target-location loss)** *Given the sigmoid function $\sigma(x) = \frac{e^x}{e^x + 1}$ and let $y_k \in \{0, 1\}$ be the outcome of the passing event at time $t(x_k)$ for a game state $x_k$, $d_k$ the destination location of the passing event $k$, $f$ a PassNet with parameters $\theta$, and $logloss$ the log-loss function, we define the target-location loss as*

$$L(f(x_k; \theta), y_k, d_k) = logloss(f(x_k; \theta)_{d_k}, y_k)$$

We approach the training of the model as weakly-supervised learning task. We consider this problem set to be an extreme case of weak learning where ground truth labels only correspond to a single location in the full mapping matrix that needs to be learned. The target-location loss presented in Definition 3.2 essentially shrinks the output of a PassNet $f$ to a single prediction value by selecting the prediction value at the destination of the pass, and then computes the log-loss between this single prediction and the ground-truth outcome value.

## 3.3 SPATIAL AND CONTEXTUAL CHANNELS FROM TRACKING DATA

Our architecture is designed to be built on top of two common sources of data for sports analytics: tracking data and event data. Tracking data consists of the location of the players and the ball at a high frequency-rate. Event-data corresponds to manually labeled observed events, such as passes, shots and goals, including the location and time of each event. We normalize the location of players and the ball to ensure the team in possession of the ball is attacking from left to right, thus standardizing the representation of the game situation. On top of players' and ball location we derive low-level input channels including spatial (location and velocity) and contextual information (ball and goal distance). An ideal choice of $(h, l)$ is the maximum permitted size of a soccer field $(105, 68)$ in meters, so every location represents $1m^2$ of space on the field. However, given that the architecture requires $1/2x$ pooling steps, we conveniently set $h$ to $104$.

**Definition 3.3 (Tracking-data snapshot)** *Let $Z_p(t), Z_d(t), Z_b(t), Z_g(t) \in \{z | z \in \mathbb{R}^{l \times h}\}$ be the locations of the attacking team players, the location of the defending team players, the location of the ball, and the location of the opponent goal, respectively, at time $t$, then a tracking-data snapshot at time $t$ is defined as the 4-tuple $Z(t) = (Z_p(t), Z_d(t), Z_b(t), Z_g(t))$.*

In order to create a game state representation $X(t) \in X$ as described in Definition 3.1 we produce 8 different channels on top of each tracking-data snapshot $Z$ where a pass has been observed. Let $(i, j) \in \mathbb{N}_{[0, l-1]} \times \mathbb{N}_{[0, h-1]}$ be a location in the coarse soccer field representation, $Z(t) = (Z_p(t), Z_d(t), Z_b(t), Z_g(t))$ the tracking-data snapshot at time $t$, $Z_k(t) \in Z(t)$, we define the channel extracting functions $C^L, C^{V_x}, C^{V_y}, C^E \in \mathbb{R}^{l \times h} \to \mathbb{R}^{l \times h}$, where $C^L(Z_k)$ produces a sparse matrix setting 1 for every $(i, j) \in Z_k$, $C^{V_x}$ and $C^{V_y}$ produce each one a sparse matrix setting the magnitude of velocity in horizontal and vertical axis, respectively, for every $(i, j) \in Z_k$, and $C^E(Z_k)$ produces a matrix with the Euclidean distance of each location to the first element $(i, j) \in Z_k$. Having this, we produce a 8-channels game-state representation such that $X = (C^L(Z_p), C^L(Z_d), C^{V_x}(Z_p), C^{V_y}(Z_p), C^{V_x}(Z_d), C^{V_y}(Z_d), C^E(Z_b), C^E(Z_g))$.

# 4 EXPERIMENTS AND RESULTS

## 4.1 DATASET

We use tracking-data and event-data from 309 English Premier League matches from the 2014/2015 season, provided by *STATS LLC*. Each match contains the $(x, y)$ location for every player and the ball sampled at 10Hz. The event-data provides the location, time, player, team and outcome for 208,419 passes. From this data we extract the channels described in Section 3.3 for a coarse $(104, 68)$ representation of a soccer field to obtain a dataset of size $208624 \times 104 \times 68 \times 8$. There are 166,791 successful passes and 41,628 missed passes, showing a highly imbalanced distribution of outcomes.

## 4.2 BENCHMARK MODELS

We compare our results against a series of benchmark models of increasing levels of complexity. We define a baseline model *Naive* that for every pass outputs the known average pass completion in the full dataset (81.54%) following a similar definition in Power et al. (2017). We build two additional models *Logistic Net* and *Dense2 Net* based on set of handcrafted features built on top of tracking-data. *Logistic Net* is a network with a single sigmoid unit, and *Dense2 Net* is a neural network with two dense layers followed *ReLu* activations and sigmoid output unit.

**Handcrafted features** We build a set of spatial features on top of tracking-data based on location and motion information on players and the ball that is similar to most of the features calculated in previous work on pass probability estimation (Power et al., 2017; Spearman et al., 2017; Gudmundsson & Horton, 2017; Hubáček et al., 2018). We define the following set of handcrafted features from each pass: origin and destination location, pass distance, attacking and defending team influence at both origin and destination, angle to goal at origin and destination, and max value of opponent influence in a straight line between origin and destination. Team's spatial influence values are calculated following the model presented in Fernández & Bornn (2018).

**Pass selection model** We provide an example of how the architecture can be extended to other similar problems by training the network to generate pass selection likelihood surfaces. This refers to the likelihood of a pass being made towards every other location on the field. To achieve this, we simply have to modify the sigmoid activation layer of the original architecture by a softmax activation layer. Softmax ensures that the sum of probabilities on the output surface add up to 1. For this case, instead of pass success, our target output is 1, indicating a pass has been made, while the loss is still computed by obtaining the log-loss between the prediction at the destination location and 1.

## 4.3 EXPERIMENTAL FRAMEWORK

**Training, validation and test sets** First the data is randomly shuffled and then split into a training, validation and test set with a $60 : 20 : 20$ distribution. We apply stratified split so the successful/missed pass class ratio remains the same across datasets. The validation set is used for model selection during a grid-search process. The test set is left as hold-out data, and results are reported on performance for this dataset. For the benchmark models, datasets are built by extracting the features described in Section 4.2, and an identical split is performed. Features are standardized column-wise by subtracting by the mean and dividing by the standard deviation.

**Approaching imbalance through domain-knowledge** We apply domain-knowledge of soccer in order to produce a balanced class outcome for the training set. We define a pass as *unreachable* if the closest player on the defending team to the destination location is $1.5$ times closer than the closest player in the attacking team. This definition was empirically validated by a group of expert soccer analysts from *F.C. Barcelona*. We randomly select successful passes without replacement and then draw random possible locations until the pass is classed as *unreachable*, and repeat this procedure until we reach a $50/50$ balanced training dataset. The validation and test dataset are left as originally sampled. A class-weighting with no upsampling approach was tested, but better results where achieved by augmentation.

**Optimization** Both the PassNet network and the baseline models are trained using adaptive moment estimation (Adam). Model selection is achieved through grid-search on learning rates of $10^{-3}$, $10^{-4}$ and $10^{-5}$, and batch sizes of 1, 16 and 32, while $\beta_1, \beta_2$ are set to 0.9 and 0.999 respectively. We use early stopping with minimum delta rate of $0.001$. Optimization is computed on a single Tesla M60 GPU and using Tensorflow 1.5.0. During the optimization the negative log-loss is minimized.

**Metrics** Let $N$ be the number of examples in the dataset, $y$ the ground-truth labels for a pass events and $\hat{y}$ the model predictions. We report the negative log-loss

$$\mathcal{L}(\hat{y}, y) = -\frac{1}{n} \sum_i y_i \cdot log(\hat{y_i}) + (1 - y_i) \cdot log(1 - \hat{y_i}).$$

In order to validate the model calibration we use a variation of the expected calibration error (ECE) presented in Guo et al. (2017) which computes the expected difference between accuracy and

confidence of the model on a finite set of samples splitted into $K$ bins of size $1/K$, according to the predicted confidence or probability for every sample. Since our model is not designed for classification we use the count of number of examples of the positive class rather than accuracy for $ECE$.

Let $B_k$ be a bin where $k \in [0, K]$ then

$$ECE = \sum_{k=1}^{K} \frac{|B_k|}{N} \left| \left( \frac{1}{|B_k|} \sum_{i \in B_k} 1(y_i = 1) \right) - \left( \frac{1}{|B_k|} \sum_{i \in B_k} \hat{y}_i \right) \right|.$$

A perfectly calibrated model will have a ECE value of 0. Additionaly we provide a calibration reliability plot (Guo et al., 2017) showing the mean confidence for every bin $B_k$.

## 4.4 RESULTS

Table 1 presents the results for the benchmark models and PassNet for the pass probability dataset. We can observe that PassNet achieves remarkably less error than the other models and produces a calibrated estimation of pass probability. Despite the considerably large number of parameters in PassNet, the inference time for a single sample is low enough to produce real time estimation for frame rates below 200Hz. The ECE results for both Logistic Net and PassNet represent a mean difference in probability estimation lower than $2.72\%$ for both models, which make them suitable for use in practice. Figure 2 presents a calibration reliability plot for each of the models. Consistently with the ECE results, both Logistic Net and PassNet produce well calibrated estimations of pass probabilities. Despite both models being calibrated, PassNet is proven to be considerable more precise when estimating individual sample probabilities, as shown by the difference in log-loss between both.

Table 1: Results for the benchmark models and PassNet for the pass probability dataset.

| Model | Log-loss | ECE | Inference time | Number of parameters |
|---|---|---|---|---|
| Naive | 0.418 | — | — | 0 |
| Logistic Net | 0.275 | **0.01854** | 0.00191s | 11 |
| Dense2 Net | 0.255 | 0.0824 | 0.00204s | 231 |
| PassNet | **0.182** | 0.0272 | 0.00457s | 397, 259 |

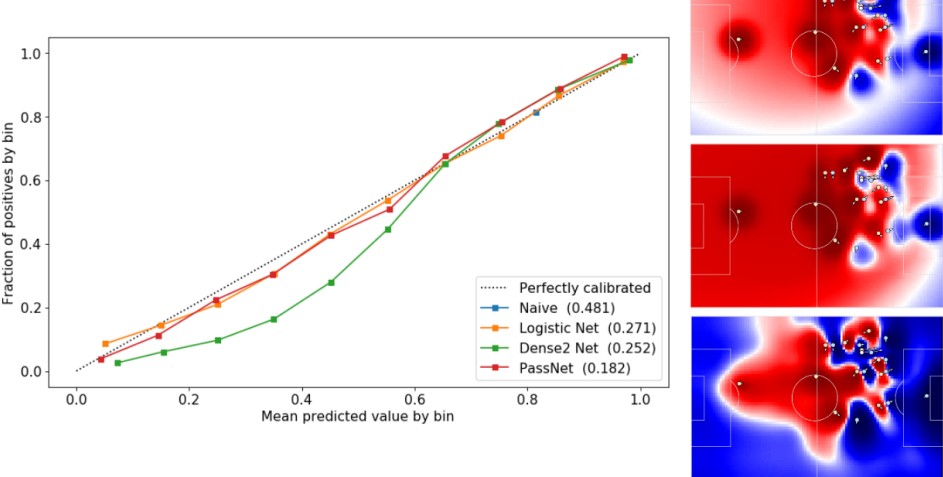

Figure 2: In the left a calibration reliability plot, where for a series of 10 bins of equal sizes, the X axis presents the mean predicted value for samples in each bin, and the Y axis the fraction of samples in each bin containing positive examples. On the right, a comparison of pass probability surfaces from Logistic Net, Dense2 Net and PassNet models, from top to bottom.

Figure 3 presents the predicted pass probability surfaces and the pass selection surface for a specific game situation during a professional soccer match (two videos showing the evolution of both surfaces in time can be obtained at `http://tiny.cc/pzcddz` and `http://tiny.cc/h1cddz`). We observe that the model is able to capture both fine-grained information, such as the influence of defending and attacking players on nearby locations, and coarse information such as the probability of reaching wider spatial areas depending on the distance to the ball and the proximity of players. We can also observe that the model considers player' speed, so it can predict probabilities of passing to not-yet occupied spaces, a critical aspect of practical soccer analysis. In the right column of Figure 2 we can also observe a comparison between predicted probabilities for the benchmark model and PassNet. The surfaces for the former are estimated by predicting the pass probability for every location individually. Although able to capture global information, both benchmark models fail to make more in-depth sense of local information, particularly when there is high density of players in local regions. They also over-estimate the probability of passing to regions that are barely reachable by any player.

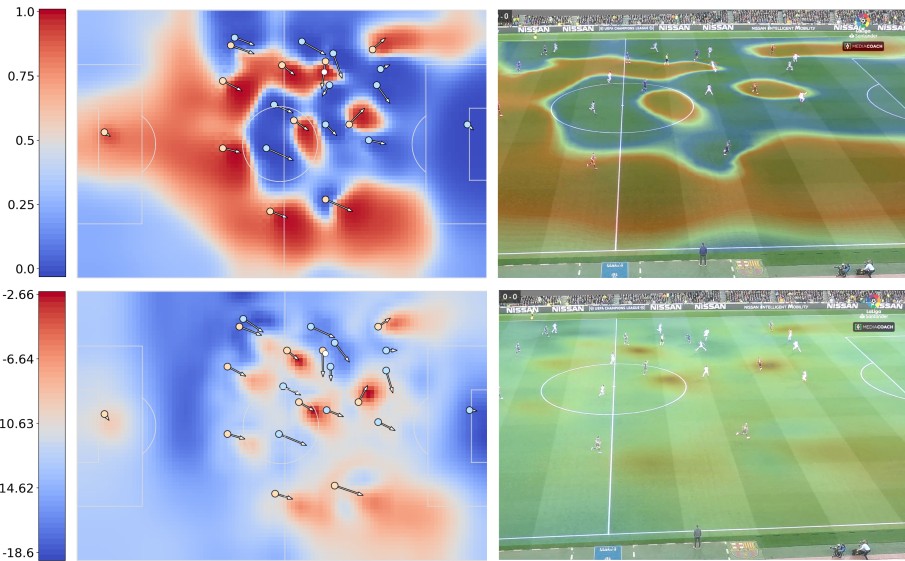

Figure 3: On the left column a 2D representation of the pass probability surface (top) and the pass selection surface (bottom) for a given game situation. Yellow and blue circles represent locations of players on the attacking and defending team, respectively, and the arrows represent the velocity vector for each player. The white circle represents the location of the ball. Pass selection likelihood is represented on logarithmic scale. The right-hand column presents a video capture from the associated soccer match with an overlay of the corresponding surface.

## 5 DISCUSSION AND FUTURE WORK

We have presented a deep neural network architecture that is able to produce full pass probability surfaces from low level spatio-temporal soccer data. The network is able to perform remarkably well at estimating the probability of observed passes, while also being able to estimate the probability of any other potential pass. By merging features extracted at different sampling levels the network is able to extract both fine and coarse details, thereby managing to make sense of the complex spatial dynamics of soccer. The features are learned following an extreme case of weakly supervised learning where only single-location labels are available and the network is forced to indirectly provide accurate predictions of individual passes by directly learning a full probability surface. The produced surfaces can provide an accurate visual representation of game situations that can be interpreted directly by soccer coaches. The network can also be adjusted to deal with many other related problems such as the estimation of pass reward, the probability of actions leading to goals, or the likelihood of ball turnover, among others. We show this by sucessfully training the same architecture to estimate surfaces of pass selection likelihood, with a few simple adjustments.

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
