# OpenReview forum: "PassNet: Learning pass probability surfaces from single-location labels. An architecture for visually-interpretable soccer analytics"
_ICLR.cc/2020/Conference — Reject_

### Official Review · AnonReviewer3 · 2019-10-18
**Official Blind Review #3**

**Rating:** 8

**Review:**

This paper presents a deep convolutional neural network for estimating, at the given the moment, for any given position on the field, the probability that a pass to that position is successful. It also shows results regarding the probability that a pass to a particular position is attempted. The paper is well-written and the figures and videos do a very nice job of communicating the important ideas. The related work appears to be extensive and the description of the design choices of the architecture and the training procedure is clear and thorough. Based on the results in Table 1, the proposed network appears to compare very favorably against the baselines.

One thing I feel would make the paper stronger is the inclusion of more baselines and an ablation study. Both Logistic Net and Dense2 Net appear to be very lightweight. It is not obvious to me that all of the design choices made in the paper translated to gains in performance. Would a large network perform well without them? You mention that you tried a class-weighting with no sampling approach but that it did perform as well. Why not include these results in the paper?

Small typo: “where achieved by augmentation” -> “were achieved by augmentation”

From the perspective of someone who is not an expert in this area, I think this is a nice paper. It appears to successfully address an interesting problem, it is well-organized, and its solution methodology may have applications to other related problems. My one major criticism is that I feel that there is insufficient information regarding how the design choices affected the performance.

**Experience Assessment:**

I do not know much about this area.

**Review Assessment: Checking Correctness Of Derivations And Theory:**

N/A

**Review Assessment: Checking Correctness Of Experiments:**

I assessed the sensibility of the experiments.

**Review Assessment: Thoroughness In Paper Reading:**

I read the paper thoroughly.

---

### Official Review · AnonReviewer1 · 2019-10-23
**Official Blind Review #1**

**Rating:** 6

**Review:**

The aim of the system presented in this paper is to produce a 2D map of probabilities showing the chance of successful completion of a soccer pass to all locations on the field, given coordinate locations of players and the ball sampled over time.  A network based on a fully-convolutional style semantic segmentation network is applied to a 2D, 8-channel game state representation, with final sigmoid layer to predict a pass success indicator at each location.  The system is trained using manually annotated pass success and destinations, which corresponds to a label on the destination point; locations other than the labeled point are not trained (treated as incomplete/"don't-care" training targets).  Evaluation is performed using both log loss and probability calibration measure, to measure effectiveness of predictions as well as how well they calibrate to correspond to probabilities in the sense defined by the measurement.

This is a fun application and it appears the system is effective at a basic level.  However, I think both the theory/explanation and experiments leave a fair bit of uncertainty as to what the probabilities mean and how to interpret them, including conflation of success probability and the model's certainty in its estimate.

In particular, it is unknown to what degree the values output by the model can be interpreted as a predictive probability of pass completion for anywhere on the field.  If one location says 0.8 and the other 0.5, does this mean that if the player were to actually pass to the 0.5 location, the chance of success is actually lower?  Or does it mean that the model is less confident or that this location was under-trained for this state?  The ECE measurement in conjunction with loss error doesn't quite address this:  although a good verification of calibration in its own sense, ECE simply confirms that for cases where the model predicts 0.5, there is success 0.5 of the time for locations that *exist in the test set*, which is sampled according to player action.  To verify the probability maps at all locations, one would need to be able to measure *any* point in the field, not just those already selected by the players' actions.  I think discussion and attempt to measure this is fairly important, as one of the intended applications in the motivation is analyzing what might have happened had a player selected a different destination point.

Unfortunately, knowing the true outcomes at arbitrary locations the players didn't pass to is impossible, so addressing this issue is not straightforward, and unclear to me how it might be done.  A possible suggestion, is to use the destination selection predictions that the paper also mentions can be found using softmax instead of sigmoid.  Although this does not entirely eliminate the issue (these predictions themselves may conflate model confidence with actual selection probability), these maps would likely provide a good indication of player selections.  Thus they might be used to sample or reweight the test set, so that unlikely destination points are sampled more, to try to get a more uniform sample.

Even with this issue, though, I feel that this is an interesting application and system that seems reasonable in its current state, if with important caveats.  Thus, I'd lean towards accepting.  However, I'd encourage the authors to discuss these differences and issues of output interpretation, and to try addressing if possible.


Additional questions and comments:

* The train/val/test split appears to be uniform random by pass event.  It would be interesting to hold out all events for one or more teams, or holding out full games, to measure generalizability to these cases.

* Are the destinations for unsuccessful passes the intercepted location, or an estimated intended location?  Does this difference affect the completion prediction at the intended location (beyond the interception location)?




**Experience Assessment:**

I have published one or two papers in this area.

**Review Assessment: Checking Correctness Of Derivations And Theory:**

I assessed the sensibility of the derivations and theory.

**Review Assessment: Checking Correctness Of Experiments:**

I assessed the sensibility of the experiments.

**Review Assessment: Thoroughness In Paper Reading:**

I read the paper at least twice and used my best judgement in assessing the paper.

---

### Official Review · AnonReviewer2 · 2019-10-23
**Official Blind Review #2**

**Rating:** 3

**Review:**

Contribution:
This paper proposes PassNet an architecture designed for soccer pass analytics. PassNet approach is similar to UNet, having a downsampling and upsampling modules with a set of skip-connection between the two modules. To train their model, authors apply the log-loss at the location of the passing event.  Authors evaluate their approach on a soccer analytic dataset, where they demonstrate improvement over prior works relying on hand-crafted features.

Comment:
Contribution of the paper appears a bit incremental to me.  It seems that the paper is a direct application of Unet type of architecture on the specific problem of pass analytic. It would be nice to explicit what is the main contribution of the paper from the representation learning point of view.

In addition, some of the design choice of PassNet could be better justified. It would be nice to run an ablation study to show the importance of the different architectural component (conv2d prediction layers, backpropagating using only one outputs, skip-connection...)

Authors claim that their approach is an "extreme case of weakly supervised learning". I tend to disagree with the assessment. I understand that they only backprop the loss computed at one specific location of the output, but this a choice on the architectural part and not on the labelling. Training PassNet requires fine-grained label as it needs to know both label value and localization. Training the model without the use of label localization would be more akin to weakly-supervised learning.

Authors evaluate their approach on only one dataset.  It would be nice to extend the empirical results to other datasets/sports to ensure the robustness of the conclusions.


**Experience Assessment:**

I have read many papers in this area.

**Review Assessment: Checking Correctness Of Derivations And Theory:**

N/A

**Review Assessment: Checking Correctness Of Experiments:**

I carefully checked the experiments.

**Review Assessment: Thoroughness In Paper Reading:**

I read the paper thoroughly.

---

### Author Response · Authors · 2019-11-13
**Authors Response to Reviews: Ablation test**

A common request was to present the results of an ablation test in order to prove the usefulness of the architecture design decisions.
The components used for the ablation test are:

Skip-connection (Yes/No)
Upsampling (Yes/No)
Fusion layer (Yes/No)
Non-linear prediction layer
Single level filters: The number of layers of convolutional filters by sampling level

+---------------------+----------------------+-----------------+-------------------+---------------------+----------------------+----------+
|        Model        | Skip-connection (SC) | Upsampling (UP) | Fusion layer (FL) | NL prediction (NLP) | Single level filters | Log-loss |
+---------------------+----------------------+-----------------+-------------------+---------------------+----------------------+----------+
|       PassNet       |          YES         |       YES       |        YES        |         YES         |           2          |   0.316  |
+---------------------+----------------------+-----------------+-------------------+---------------------+----------------------+----------+
|     PassNet-NLP     |          YES         |       YES       |        YES        |          NO         |           2          |   0.353  |
+---------------------+----------------------+-----------------+-------------------+---------------------+----------------------+----------+
|      PassNet-FL     |          YES         |       YES       |         NO        |         YES         |           2          |   0.290  |
+---------------------+----------------------+-----------------+-------------------+---------------------+----------------------+----------+
|    PassNet-FL-NLP   |          YES         |       YES       |         NO        |          NO         |           2          |   0.421  |
+---------------------+----------------------+-----------------+-------------------+---------------------+----------------------+----------+
|      PassNet-UP     |          YES         |        NO       |        YES        |         YES         |           2          |   0.315  |
+---------------------+----------------------+-----------------+-------------------+---------------------+----------------------+----------+
|    PassNet-UP-FL    |          YES         |        NO       |         NO        |         YES         |           2          |   0.317  |
+---------------------+----------------------+-----------------+-------------------+---------------------+----------------------+----------+
|    PassNet-UP-NLP   |          YES         |        NO       |        YES        |          NO         |           2          |   0.323  |
+---------------------+----------------------+-----------------+-------------------+---------------------+----------------------+----------+
|  PassNet-UP-FL-NLP  |          YES         |        NO       |         NO        |          NO         |           2          |   0.335  |
+---------------------+----------------------+-----------------+-------------------+---------------------+----------------------+----------+
| Single Layer CNN-D4 |          NO          |       YES       |        YES        |         YES         |           2          |   0.364  |
+---------------------+----------------------+-----------------+-------------------+---------------------+----------------------+----------+
| Single Layer CNN-D8 |          NO          |       YES       |        YES        |         YES         |           4          |   0.320  |
+---------------------+----------------------+-----------------+-------------------+---------------------+----------------------+----------+
PassNet achieved the best log-loss together with PassNet-FL and PassNet-UP. When performing a visual evaluation of the quality of the generated surfaces we can see that there are a few subtle differences between PasseNet-FL (see image https://ibb.co/WVpDq38 ) and PassNet (see image https://ibb.co/RNjj07s ), but they can be considered largely equivalent. Given that the best loss is achieved by PassNet-FL we could agree that the fusion layer could be removed from the architecture.

Additionally, PassNet-UP-FL performs similarly to PassNet in terms of log-loss. However, if we take a closer look at the generated surface (see image https://ibb.co/h1QqpDt ) we can see that the output is artifacted, while PassNet surface provides an eye-pleasing smooth surface. These smooth surfaces are preferable in practice for better communication with coaches. Given the marginal difference between both it is preferable to keep the upsampling layers.

Single Layer architectures perform similarly to PassNet according to log-loss, however if we take a look at the generated surface we can see that the net overfits and predicts exclusively 0.5 probabilities near dense opponents areas (see image https://ibb.co/hWHrDFd ).

From this analysis we can conclude that all of the proposed components (with an exception of fusion layers) provide value to the model.

---

### Author Response · Authors · 2019-11-13
**Authors Response to Reviews: Testing on an additional hold-out dataset:**

We would like to thank all the reviewers for their useful feedback and are glad to see that the paper was well received by most.
Below we provide answers to the main questions sent by the reviewers.

Recently we have obtained tracking data for an additional full season of the English Premier League (13/14).
The dataset includes 237,128 passes from matches that have not been previously fed to PassNet during the development of the paper. Below we present the logloss and excepted calibration error (ECE) between the proposed PassNet architecture and the baseline models. We can observe that PassNet is still considerably better than the baseline models in this new test set.

+--------------+----------+-------+
|     Model    | Log-loss |  ECE  |
+--------------+----------+-------+
|     Naive    |   0.488  |   -   |
+--------------+----------+-------+
| Logistic Net |   0.510  | 0.117 |
+--------------+----------+-------+
|  Dense2 Net  |   0.390  | 0.130 |
+--------------+----------+-------+
|    PassNet   |   0.316  | 0.063 |
+--------------+----------+-------+

---

### Author Response · Authors · 2019-11-13
**Authors Response to Reviews: Evaluating unseen passes and accounting for unlike passes**

One of the reviewers commented that the only way to objectively evaluate probability prediction at every other location different from the observed pass destination location is to compare with the full probability surface as ground-truth, which is impossible in practice. The reviewer suggests to use the pass likelihood model that can be obtained by substituting the sigmoid activation output layer by a softmax and use that pass likelihood to evaluate different types of situations, that might be undersampled in the original data.
In order to test that the model performs robustly on less likely passes, so we can provide a higher confidence level of the quality of the predicted surfaces, we performed the following test: for all the passes in the new test dataset we predicted the likelihood of observed passes and we applied K-means with K=5 to obtain three incremental pass likelihood groups, named according the likelihood of the passing location within the group: very low, low, medium, high, and very high.

The table below shows the the likelihood ranges (for a 104x68 grid) and the log loss between the predicted probability in each group and the observed outcome for the PassNet architecture.


+-----------------+-----------+-----------+------------------+
| Pass Likelihood | Min Value | Max Value | PassNet log-loss |
+-----------------+-----------+-----------+------------------+
|     Very low    |   1e-10   |   0.007   |       0.543      |
+-----------------+-----------+-----------+------------------+
|       Low       |   0.007   |   0.019   |       0.134      |
+-----------------+-----------+-----------+------------------+
|      Medium     |   0.019   |   0.033   |       0.083      |
+-----------------+-----------+-----------+------------------+
|       High      |   0.033   |   0.052   |       0.061      |
+-----------------+-----------+-----------+------------------+
|    Very high    |   0.052   |   0.136   |       0.049      |
+-----------------+-----------+-----------+------------------+

We can clearly observe that the more likely a passing destination location is the better the prediction of PassNet. Despite a worst than average prediction of
very unlikely passing locations, PassNet is able to perform considerably well in the rest of the cases. We expect the network to suffer from the noise introduced by the labels of destination locations, where in case of failed passes it is likely the data provider tags the interception location of the pass instead of the intended location. In future work we can address this problem by resampling the training set according to the unlikely passes ratio observed in the data, as well as developing an intended receiver prediction model.

---

### Author Response · Authors · 2019-11-13
**Authors Response to Reviews: What is the value of PassNet in terms of representation learning**

While the use of skip-connections and some other of the components of this network can be found in several of the cited previous work in image segmentation, we consider PassNet provides a set of novel additions to this field. First, this is to our knowledge the first approach of this kind applied in sports analytics, where a full surface prediction is provided from full resolution tracking data. Also, while these kind of architectures have been proved successful when either a full ground-truth map correspondance is available or a single label related to objects in the input image is provided, we prove that high level features can be learn from just single-pixel level ground-truth. Also, this paper shows how complex representations can be learned through a large model (in parameters size compared to baseline models) while also providing rich visual interpretation that can be directly applied in practice. Also the network is not limited to estimating passing probabilities but we show that with few modifications it can be adjusted to learn many other similar problems in this field, allowing as well for a possible application in other sports where tracking data is also available.

---

### Comment · Area_Chair1 · 2019-11-13
**Thanks for your reviews. Please take a look at the rebuttal.**

Dear reviewers,

Thank you very much for your efforts in reviewing this paper.

The authors have provided their rebuttal. It would be great if you take a look at them, and see whether it changes your opinion in anyway. If there is still any unclear point or a serious disagreement, please bring it up. Also if you are hoping to see a specific change or clarification in the paper before you update your score, please mention it.

The authors have only until November 15th to reply back.

I also encourage you to take a look at each others’ reviews. There might be a remark in other reviews that changes your opinion.

Thank you,
Area Chair

---

### Decision · Program_Chairs · 2019-12-19

**Decision:**

Reject

**Comment:**

The paper proposes PassNet, which is an architecture that produces a 2D map of probability of successful completion of a soccer pass. The architecture has some similarities with UNet and has downsampling and upsampling modules with a set of skip-connections between them.

The reviewers raised several issues:
* Novelty compared to UNET
* Lack of ablation studies
* Uncertainty about what probabilities mean and issues regarding output interpretation.

The authors have tried to address these concerns in their rebuttal and provided additional experiments. They also argue that the application area (sport analytics) of the paper is novel. Even though the application area is interesting and might lead to new problems, this paper did not get enough support from reviewers to justify its acceptance.